# The Correlates & Public Health Consequences of Prospective Vaccine Hesitancy among Individuals Who Received COVID-19 Vaccine Boosters in the U.S.

**DOI:** 10.3390/vaccines10111791

**Published:** 2022-10-25

**Authors:** Matt Motta

**Affiliations:** Department of Health Law, Policy & Management, Boston University School of Public Health, Boston, MA 02118, USA; mmotta@bu.edu

**Keywords:** COVID-19, vaccine hesitancy, public opinion, booster shots, health attitudes, health behavior

## Abstract

Supplemental “booster” vaccines may prove vital in combating variant waves of endemic COVID-19. Given relatively low levels of booster vaccine uptake, Americans’ willingness to receive a *second* booster shot is unclear. In a demographically representative survey of N = 3950 US adults (limited to a “boosted” subsample of N = 1551 who had not yet received a second booster), 49% [95% CI: 47, 51] of Americans report having received an initial booster shot, while just 34% [33, 36] report that they would be “very likely” to do so again. Concerns about missing work to vaccinate (−10%; B = 0.53, *p* = 0.05) and being unconvinced that additional boosters will be necessary (−47%; B = 2.24, *p* < 0.01) are significantly and negatively associated with being very likely to receive a second COVID-19 booster. These findings can help inform discussions about policies aimed at increasing vaccine uptake in the U.S., and broaden researchers’ understanding of vaccine reluctance among those who might otherwise hold positive views toward vaccination.

## 1. Introduction

As of April 2022, less than half of Americans who are fully vaccinated against COVID-19 (i.e., those who received two doses of Pfizer or Moderna mRNA vaccines, or one dose of the Johnson & Johnson vaccine) opted to receive an additional “booster” shot. Low levels of booster vaccine uptake could pose an important public health concern in the U.S., as several studies have documented waning vaccine effectiveness over time [1,2]; both due to a decreased immune response, and the emergence of new viral variants. As a result, even vaccinated Americans may be at risk of contracting new COVID-19 infections, or spreading the disease to others, including vulnerable unvaccinated populations. 

Troublingly, even booster shots are no guarantee of lasting immunity, as the efficacy of initial booster vaccines has *also* been shown to wane over time [3]. Consequently, federal regulators at the Food and Drug Administration (FDA) have authorized individuals most at risk of experiencing severe complications from COVID-19 to receive an additional booster shot; including those over the age of 50 and/or who are immunocompromised [4]. Moreover, recognizing the problem of waning immunity for even “boosted” Americans resulting from the rise of the BA.5 Omicron variant [5,6,7], the Food and Drug Administration (FDA) considered expanding booster eligibility to all Americans in late July, 2022 [8]. Ultimately, the FDA opted to delay recommending widespread booster vaccination in Summer 2022, in order to encourage all adults to receive (at the time) new Omicron-adjusted booster vaccines (which, in theory, would provide greater immunity to novel variant infections) [8], anticipated to be available in Fall 2022.

Thus, while vaccination offers both increased COVID-19 immunity and protection from severe infection [9], these effects may be temporary as new viral variants emerge. Correspondingly, Americans may need to receive booster vaccines regularly in order to reduce the chance of becoming infected with COVID-19. Efforts to study the American public’s willingness to receive an additional booster shot are therefore an important area for research. If and when new viral variants emerge, vaccines—along with antiviral medication and pro-social health behavior (e.g., social distancing, masking)—may play a role in not only limiting serious illness, but slowing the spread of viral infections. 

Presently, Americans’ willingness to receive a *second* booster vaccine—as well as the social and health-related motivations underlying their willingness to do so—remain an open question. Critical for understanding this phenomenon is recognizing that vaccine hesitancy, unlike indicators of whether or not people are vaccinated, is not a binary construct. Choosing to vaccinate does not necessarily imply the absence of vaccine hesitancy [10,11]. Recent work shows, for example, that even those who have chosen to vaccinate against COVID-19 [12] in the past may have done so despite expressing some reservations about their decision, and might plausibly be considered to be “vaccine hesitant.” Relatedly, vaccine hesitancy may also be dynamic if individuals update their vaccine-related attitudes following particularly positive or negative past vaccination experiences, e.g., if the experience of mild fever or chills post-vaccination engenders elevated concerns about vaccine side effects.

Correspondingly, there are at least four reasons to suspect that some Americans who opted to receive a (first) booster shot may nevertheless forego receiving a second (“prospective vaccine hesitancy,” or “PVH”).

First, some Americans who opted to receive their first COVID-19 vaccine booster may be reluctant to do so a second time due to fears about the possibility of experiencing mild side effects. Although vaccine boosters are not thought to increase the likelihood that vaccinated individuals experience severe side effects (e.g., those requiring hospitalization), Americans have been significantly more likely to report experiencing more-mild side effects (e.g., fever, chills) following booster vaccine administration [13]. Boosted individuals who experienced these symptoms following their booster shot, who are attentive to anecdotal coverage of side effect concerns in the popular press, or who know someone who experienced side effects may therefore be more reluctant to vaccinate a second time.

Second, and very much relatedly, some boosted individuals may opt to forego vaccinating a second time due to fears about missing work from either vaccine administration or the experience of mild side effects. At the outset of COVID-19 vaccine administration in Spring 2021, public opinion research from the Kaiser Family Foundation reported that nearly half of Americans (48%)—including 42% of those most-eager to receive a COVID-19 vaccine—worried that they might miss work in order to receive the vaccine and/or following the experience of mild side effects [14]. These concerns were understandable, given that (in the same survey) just 23% reported that their employer offered to give them time off in order to receive the vaccine or recover from any side effects they might experience. 

Correspondingly, fears about missing work offer a unique insight into COVID-19 vaccine hesitancy. Americans who express this view are not necessarily *opposed* to receiving a second booster shot, but may nevertheless ultimately opt to forego doing so, due to fears that they may miss time at work in order to receive the vaccine, or to deal with its potential side effects.

Third, some individuals may not be convinced that receiving an additional booster shot could be necessary, even if government regulators authorize it as such. While public concern about the public health consequences of COVID-19 has tended to grow in response to elevated transmission attributable to new viral variants, these considerations tend to be short-lived [15]. Correspondingly, Americans may have difficulty envisioning a future wherein receiving additional booster shots might be necessary to curb viral transmission, especially in periods in-between major disease outbreak waves. Some may also view vaccination as unnecessary as more Americans become infected with (and develop antibodies to) COVID-19; even though these protections, like vaccines, also wane over time, and are likely temporary due to the emergence of new viral variants [16].

Finally, some might express altruistic concerns about the “zero sum” nature of booster vaccine administration in the US. Unlike the original vaccine regimen and first booster shot, stalled budgetary legislation in Congress in Spring 2022 (i.e., when the present research was conducted. See: Methods) caused the Biden Administration to fall short of funding goals necessary to purchase enough vaccine doses for all Americans who might want to receive a shot [17]. As of Summer 2022, this measure had yet to receive Congressional funding; thereby jeopardizing both present vaccine availability and future vaccine supply [18]. Correspondingly, some Americans who are less at risk of developing severe illness from COVID-19 may worry that choosing to receive a second booster might deprive someone in greater need of vaccination from having access to the shot.

Presently, the extent to which Americans who opted to receive a (first) booster shot might be willing to receive a second is unclear. Moreover, few have investigated the degree to which each of the above factors might influence reluctance to receive a second booster shot, if and when one is authorized. 

At first, it may seem intuitive, and perhaps even uninteresting, to suggest that Americans who received a first booster shot may share the general public’s concerns about vaccine side effects, the possibility of missing work, or the “zero sum” consequences of receiving a second booster; thereby leading them to forego future vaccination. Yet, it is important to bear in mind that individuals who received the first booster either did so *despite* these concerns, and/or have become more concerned about these topics in recent months. 

The minority of Americans who have received a COVID-19 booster shot are among the most vaccine-accepting members of the population, as evidenced by their willingness to seek out an additional (and, for most, optional) vaccine. If substantial levels of prospective vaccine hesitancy (PVH) regarding future COVID-19 vaccination exist among even this comparatively vaccine-accepting group, an important element of our public health response to future pandemic waves may be weakened. 

Consequently, my goal in this research is to ask (Research Question 1: RQ1) whether or not those who have already received one vaccine booster might be willing to do so again, and (RQ2) why some of these individuals might forego receiving vaccination. 

## 2. Methods

### 2.1. Data 

Data for this study are derived from a demographically representative sample of N = 3950 US adults, conducted between 22–27 April 2022 via Lucid Theorem. Lucid invited 4155 individuals to participate in this study, and 205 did not consent to complete the survey. This is indicative of a 95% compliance rate.

Lucid Theorem is an online opt-in sampling service that retains a large panel of potential survey respondents (over 375,000 unique daily users are available for contact, by some estimates). [19] The service invites respondents to participate in surveys using quota sampling methods. This means that, based on responses to an initial demographic inventory survey, the service selectively sends survey-taking opportunities to potential respondents who fill quotas derived from nationally representative demographic benchmarks obtained from the US Census.

Note that, as this study pertains to the prevalence and correlates of vaccine hesitancy among those who opted to receive a booster, analysis is limited to the N = 1551 individuals who self-reported receiving a booster shot (and who had not yet received a second booster). Question wordings for a series of branched questions used to implement this “filter” are available in the Appendix A, and are summarized in Table 1.

Although data from Lucid are not formally nationally representative, the service’s quota sampling procedures have been shown to build samples that are demographically representative of US adults with respect to respondents’ age, gender identity, income, racial identity, political party, and geographic region [19]. Correspondingly, data from Lucid have been shown to produce estimates of vaccine hesitancy that mirror those obtained in nationally representative probability samples [20]. Data from Lucid have also been shown to replicate well-studied experimental treatment effects both prior to and throughout the COVID-19 pandemic [21]; tend to better reflect US population demographic benchmarks than traditional convenience samples [22]; and have been employed extensively in the study of COVID-19 vaccine attitudes and behaviors in the US [19,23,24,25]. Correspondingly, although the sampling service is neither formally representative of the US population nor (necessarily) the subgroup of “boosted” Americans referenced above, Lucid’s ability to produce large and demographically representative samples (on the aforementioned factors)—coupled with its widespread usage and validation in the study of vaccine attitudes and behavior—nevertheless make it well-suited for studying the research questions tested in this piece.

To account for any potential remaining deviations between the resulting sample and the US population, post-stratification weights that adjust the data to Census benchmarks—such that under-represented responses relative to national benchmarks are treated as more influential (“up-weighted”) and over-represented respondents are treated as less influential (“down-weighted”) in the estimation of weighted means—were calculated on respondents’ age, income, gender identity, and racial identity [20]. 

Note also that, for analytical completeness, all results are presented both with and without the application of survey weights. A comparison of weighted and unweighted sample statistics to nationally representative benchmarks is available in the Appendix A. Additionally, all respondents were prevented from leaving questions blank in the survey without first reading a short (automatically generated) reminder to fill out unanswered questions. Correspondingly, there was no missing data on any of the variables featured in Table 1.

### 2.2. Measures

The primary outcome variable in this analysis is a measure of whether or not the N = 1551 respondents who opted to receive a first booster shot (and had not yet received a second) intend to receive a second booster vaccine if and when one is approved for public use. Specifically, respondents were introduced to the possibility of receiving an additional booster shot by providing them with the following information:


*As you may know, public health experts now recommend that all adults receive an additional “booster” dose of the COVID-19 vaccine. They have also authorized additional (second) boosters for vulnerable populations.*


Immediately following the presentation of this text, respondents who received a first booster shot were asked:


*If, in the future, federal regulators were to recommend that all Americans receive an additional (second) booster shot, in order to reduce the likelihood of becoming sick with COVID-19 and/or spreading it to others, how likely would you be to take the shot?*


Respondents could indicate that they were “very likely,” “somewhat likely,” “not too likely,” or “not likely at all” at all to receive a second booster. Additionally, because my aim in this paper is to study prospective booster vaccine hesitancy among those yet to vaccinate, individuals who already received a second booster shot (i.e., those aged 50 or older or those younger than 50 who are immunocompromised) could indicate that they had already received a second booster shot. After excluding from analysis those who already received a second booster shot, the resulting vaccine hesitancy was scored to scale to range from 1 (very likely) to 4 (not likely at all). 

The primary explanatory variables in this analysis are indicators of the reasons respondents provided for potentially refusing a second booster shot. Specifically, respondents were asked: 


*Which of the following concerns (if any) best describe(s) why you might be unlikely to receive an additional booster shot? Please select all that apply.*


Available response options (and both their raw and weighted sub-sample frequencies) are summarized in Table 1 (see: “Explanations for Vaccine Hesitancy”).

Additionally, models control for a host of socio-political and demographic factors that have been shown to be correlated with COVID-19 vaccine hesitancy in the US [23,26], including respondents’ political ideology (a standard seven-point scale ranging from “extremely liberal” to “extremely conservative;” [27], anti-expert attitudes (a seven-point Likert scale—ranging from “strongly agree” to “strongly disagree”—measuring agreement with the statement that respondents would “rather put [their] trust in the wisdom of ordinary people than the opinions of experts and intellectuals;” [28,29,30], gender (a dichotomous indicator of whether or not respondents self-identify as women), racial identity (dichotomous indicators of whether or not respondents self-identify as Black or Hispanic), age (dichotomous indicators of whether respondents are aged 18–24, 25–44, 45–64, or 65+), and educational attainment (a dichotomous indicator of whether or not respondents completed college). For consistency, all variables were recoded to range from 0–1.

## 3. Results

Table 1 begins by offering a descriptive assessment of vaccine attitudes and behavior in the sample. For brevity, this analysis focuses only on the non-weighted averages presented in Table 1 (although, as the table makes clear, the results are quite similar using both approaches). 

A majority (72% [70, 73]) of Americans self-report having been fully vaccinated against COVID-19. An additional 49% [47, 51] self-report having received a booster shot, with 11% [10, 12] of the sample (included among the 49% who received a first booster) indicating that they have already received an additional second booster. These figures closely resemble CDC vaccination benchmarks (CDC 2022), which (as of this writing) suggest that 77% of adults are fully vaccinated against COVID-19, with 48% receiving at least a first booster dose.

Next, Table 1 offers both a univariate (RQ1) and multivariate (RQ2) analysis of the reasons why some boosted Americans are reluctant to do so a second time. Among the N = 1551 respondents who reported receiving a first (but not second) booster dose, just 34% [33, 36] report that they would be “very likely” to do so a second time (RQ1).

Table 1 also offers initial insights into why some booster Americans may be reluctant to receive a second shot (RQ2), suggesting that more approximately a quarter expressed concerns about side effect severity (25% [22, 27]), or were unconvinced that a second booster would be necessary (25% [23, 27]).

Table 2 expands on these analyses by presenting the results of the previously mentioned multivariate ordered logistic regression models. Parameter estimates are listed at the top of each cell, with standard errors in parentheses. The results presented in the table suggest that—controlling for a variety of other factors thought to be associated with COVID-19 vaccine hesitancy—only concern about missing work to get vaccinated (B = 0.54, *p* = 0.05; *two-tailed*) and lack of concern about the need for a second booster (B = 2.25, *p* < 0.01) are significantly associated with intentions to refuse a second booster vaccine. 

Table 2 also suggests that Americans’ side-effect-related or “zero-sum” concerns about vaccination are not significantly associated with vaccination intentions. On the one hand, this is somewhat surprising, given the national trends cited earlier (i.e., suggesting that these considerations loom large for the American public, in general). However, given that my analyses focus *specifically* on those who were willing and/or able to receive a first booster shot, it may be the case that self-selection into the sub-sample can help explain why these concerns are not more prominent; i.e., because these are people who had the motivation or the resources to be able to receive a first booster, and who were evidently not deterred by the possibility of experiencing side effects following their initial vaccination.

Recognizing that ordered logistic parameter estimates are not readily interpretable on their own, Figure 1 uses the information presented in Table 2 to calculate the predicted probability that respondents are “very likely” to receive a second booster. Probing just the statistically significant terms, and holding all other covariates at their sample means, the predicted probabilities presented in Figure 1 suggest that concerns about missing work to get vaccinated is associated with a 10 percentage point decrease in the predicted likelihood that respondents are “very likely” to plan to receive a second booster (from 66 to 56%); a difference which is statistically significant at the *p* < 0.05 level, two-tailed (t = −2.03; *p* = 0.04). Similarly, albeit at a much greater magnitude, a lack of concern about the need to boost a second time is associated with a 47% decrease in the likelihood of being “very likely” to do so (from 77 to 30%). These quantities too are significantly different from one another at the *p* < 0.05 level, two-tailed (t = −22.62, *p* < 0.01). 

Collectively, these results suggest that individuals who fear missing work and/or are not convinced that additional “boosting” will be necessary are significantly less likely to indicate that they would be “very likely” to receive an additional booster shot. 

## 4. Discussion

The results presented in this paper suggest that, even among those Americans who chose to receive a booster vaccine for COVID-19, many would express substantial reluctance to do so a second time if recommended to do so by public health authorities. Concerns about the possibility of missing work to get vaccinated, along with being unconvinced that additional booster vaccinations may be necessary, play a key role in explaining why even some of the most vaccine-accepting members of the population might be unwilling to receive a second COVID-19 booster shot.

These findings have important public health implications. Vaccinating against COVID-19 will likely play an important role in curbing viral transmission if and when additional variants emerge. Amid a context of waning vaccine effectiveness and the possibility of future viral variants, and already-low levels of (initial) booster vaccine uptake, reluctance to vaccinate among even those Americans who opted to receive a first booster shot could present a substantial roadblock to curbing viral transmission. 

More-optimistically, however, documenting the potential causes of vaccine hesitancy among boosted populations could prove critical in encouraging future vaccine uptake. Per the findings presented in this paper, vaccine communicators might—if authorized as such by the FDA—choose to highlight that while some boosted individuals may experience side effects, they tend to be both short lived and mild. Moreover, health policy advocates should make a significant push to encourage the enactment of policies that incentivize employers to either compensate (or, otherwise, not punish) employees who choose to receive a second booster shot. Both of these measures could prove critical in encouraging vaccine uptake, if and when the time comes.

These findings have potentially important clinical implications as well. As this research suggests, booster vaccine uptake is both a medical decision and a social act. Healthcare providers’ conversations with their patients should therefore not only emphasize COVID-19 vaccine safety, but highlight the benefits of vaccinating despite uncertainty about its necessity and/or concerns about potentially missing work and other activities due to the experience of mild side effects.

Of course, these findings are not without limitations. First, although past research suggests that vaccination attitudes and intentions tend to strongly predict actual behavior [31,32,33], prospective behavioral self-reports do not necessarily guarantee that individuals will (or will not) choose to vaccinate. Moreover, these data are derived from a single online, opt-in cross-sectional survey. Although vaccine attitudes measured in Lucid data have been shown to closely mirror more-representative surveys [20], panelists may hold attitudes or express behaviors that differ from the US population in potentially unobserved ways. Efforts to replicate this work in formally representative data and/or to validate vaccination self-reports with verified medical records thereby offer an important avenue for future research.

Finally, this study is necessarily limited in substantive scope. For example, due to data limitations, this research cannot assess the effects of the recency and/or severity COVID-19 infections on booster vaccine attitudes. In the future, scholars ought to consider (for example) whether or not views about the public health necessity of additional boosters are influenced by personal experiences with COVID-19 infection (e.g., the idea they impart lasting immunity), fears about long term side effects from contracting the disease (“long COVID”), and considerations regarding the possibility of receiving influenza and COVID-19 booster shots in a single trip to the pharmacy or doctor’s office. More generally, researchers should continue to assess public opinion on booster-related issues, investigate correlates of booster vaccine uptake that go beyond those reported in this paper (e.g., health insurance status; as vaccine access may become more limited in the coming months), and consider conducting longitudinal studies that facilitate tracking changes in behavioral intentions (and their potential causes) over time.

## Figures and Tables

**Figure 1 vaccines-10-01791-f001:**
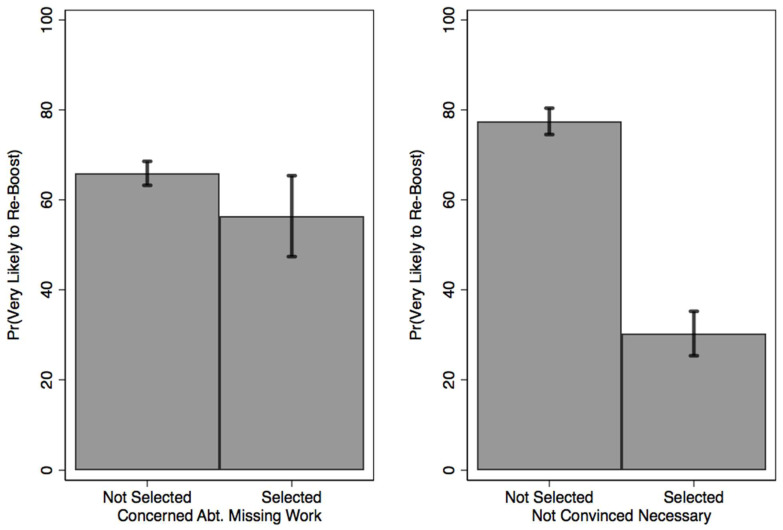
The Effect of Work-Related Concerns & Public Health Considerations on Vaccine Uptake. Note. Predicted probabilities (bars) are derived from logistic regression models that hold all covariates at their sample means. Predicted probabilities indicate the likelihood that respondents are “very likely” to intend to receive an additional booster vaccine are presented. Ninety-five percent confidence intervals extend out from each prediction. Analyses are limited to the sub-population who received a first (but not second) booster vaccine.

**Table 1 vaccines-10-01791-t001:** Raw and Weighted Summary Statistics: Vaccine Attitudes and Reported/Intended Behaviors.

	Unadjusted (Raw)	Adjusted (Weighted)
**Vaccination Behavior**		
% who are fully vaccinated against COVID-19	72 [70, 73]	74 [72, 75]
% who have received a first booster shot	49 [47, 51]	52 [50, 54]
% who already received a second booster shot	11 [10, 12]	12 [11, 14]
*Resulting Valid Subsample Size (Row 2, Excluding Row 3)*	*N = 1551*	
**Vaccination Intentions**		
% who are “very likely” to intent to receive a second booster shot	34 [33, 36]	34 [32, 36]
**Explanations for Prospective Vaccine Hesitancy (Boosted Sub-Population)**		
“I worry about the potential severity of post-vaccination side effects”	25 [22, 27]	25 [22, 27]
“I worry about missing work to get the vaccine”	9 [8, 11]	10 [8, 11]
“I worry about missing work due to side effects”	12 [10, 14]	12 [10, 14]
“I just don’t have the time”	9 [7, 10]	10 [8, 12]
“I am not convinced that another dose will be necessary (even if recommended by health experts)”	25 [23, 27]	26 [24, 29]
“I worry that receiving an additional booster will prevent others who might need it from getting vaccinated”	13 [11, 14]	12 [10, 14]

Note. Adjusted averages are calculated via the application of post-stratification weights. Please see the Appendix A for a comparison of weighted and unweighted sample demographics to nationally representative benchmarks.

**Table 2 vaccines-10-01791-t002:** The Effect of Concerns about Booster Vaccination on Vaccine Uptake Intentions for Americans who Received an Initial Booster Vaccine.

Dependent Variable = Intention to Receive an Additional Booster Shot *1 = “Very Likely;” 2 = “Somewhat Likely;” 3 “Not too Likely”; 4 = “Not Likely at All”*	
Concern: Side Effects	0.11 (0.19)
Concern: Miss Work to Vax	0.54 * (0.27)
Concern: Miss Work due to Side Effects	−0.32 (0.23)
Concern: No Time	0.51 (0.27)
Concern: Not Convinced Will Be Necessary	2.25 * (0.16)
Concern: Zero-Sum Considerations	−0.43 (0.24)
Anti-Expert Attitudes	−0.05 (0.25)
Ideology: Conservatism	1.33 * (0.25)
Gender Self-ID: Female	0.18 (0.15)
Race Self-ID: Black	−0.03 (0.23)
Race Self-ID: Hispanic	0.15 (0.27)
Age: 25–44	−0.90 * (0.28)
Age: 44–65	−0.91 * (0.29)
Age: 65+	−1.65 * (0.31)
**τ**1	1.02 * (0.31)
**τ**2	2.83 * (0.33)
**τ**3	4.32 * (0.37)
N	1528

* *p* < 0.05, two-tailed. Note. Ordered Logistic regression parameter estimates presented, with standard errors in parentheses. Please see the Appendix A for a full list of question wording information. Analyses are limited to those who received an initial, but not a second, booster vaccine.

## Data Availability

The datasets generated during and analyzed during the current study are available in the Open Science Foundation repository, at https://osf.io/rd24m/ (accessed on 23 October 2022).

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
