# Peer review of "The Correlates & Public Health Consequences of Prospective Vaccine Hesitancy among Individuals Who Received COVID-19 Vaccine Boosters in the U.S."

_vaccines, 2022, doi:10.3390/vaccines10111791_

Round 1
Reviewer 1 Report
This is an interesting and important research topic. The introduction of the background and context was clearly presented. Study is well presented and written with no major concerns in terms of the methods used and data analysis.
Two comments are suggested: 1) Based upon the findings and the limitations presented, the authors could have included suggestions on future direction and variables to be included in further study. 2) Minor revision is suggested to amend the formatting / spelling errors.
Author Response
REVIEWER 1
This is an interesting and important research topic. The introduction of the background and context was clearly presented. Study is well presented and written with no major concerns in terms of the methods used and data analysis.
I thank R1 for their kind and positive remarks about this manuscript, and again want to note that
I appreciate the reviewer’s willingness to take the time to consider this piece for publication.
Two comments are suggested: 1) Based upon the findings and the limitations presented, the authors could have included suggestions on future direction and variables to be included in further study. 2) Minor revision is suggested to amend the formatting / spelling errors.
I have made a series of mostly small changes to the manuscript in response to R1’s request for
minor revisions. First, I have added language to the piece’s discussion section that more-fully
details potential avenues for future research. This includes the identification of several potential
covariates and/or moderators of the effects documented in this piece; including how emerging
concerns about “Long COVID” might influence vaccine uptake, as well as how access to health
insurance might shape vaccination in the likely event that federal funding for US vaccination
programs expires.
Additionally, in response to R2’s helpful suggestions, I have made many small edits to the piece’s
grammar, syntax, and style. While I hope to have caught all typos and instances of unclear
wording, I am of course happy to continue to make changes throughout the revisions process.
Reviewer 2 Report
This manuscript presents a study on the hesitancy in receiving a 2nd booster vaccination for COVID in individuals after a 1st booster vaccination. The study was conducted in the USA. The author concludes in the first sentence of the Discussion “The results presented in this paper suggest that, even among those Americans who chose to receive a booster vaccine for COVID-19, many would express substantial reluctance to do so a second time if recommended to do so by public health authorities.”
The topic of this manuscript is indeed quite relevant for a separate study, even considering the intense discussion in the USA about COVID vaccination. However, the report of this study manifests serious flaws, both in Materials and Methods, and in the Results. Therefore, it is not possible to have a clear insight in the study design and hence the results, it is not possible to the reader to understand the basis on which the conclusion is based. Some points are mentioned below. A serious rephrasing of the report is indicated.
· This study is limited to the situation in the US: this should be mentioned in the title and emphasized in the introduction. For instance, the study addresses the 2nd booster which could include components of the Omicron variant, while in Europe the campaign for a 3rd booster including this variant has been initiated at the same time.
· The title needs anyhow a rephrasing: the study does not address “health policy consequences”, and “social correlates” is a very vague term which raises more questions than answers.
· The author uses the wording “social scientist”. This creates confusion, is the a scientist who is member of a socialist party, or a scientist who considers him/herself a socialist?
· The phrasings starting with “I”, like I tested, I choose, I analysed, etc, should be largely avoided, as it insinuates personal opinions instead of a science-based approach.
· The introduction is quite long, and lacks relevant information used in vaccination campaigns. For instance, booster vaccination for older people is to protect themselves, while for younger people it is to protect the society. Also, the main aim is to avoid hospitalization, and not to get infected with moderate illness.
· Line 145 “Lucid Theorem” should be explained. Apparently individuals selected in this way were approached with questions. The total of 3950 individuals were limited to 1551 individuals who self reported a booster vaccination (line 147). This quite unclear, was the study in 1551 or 3950 individuals? Where there people who did not give answers to the questions? See line 153 “Immediately following the presentation of this text, I asked respondents:”. It is assumed that non-repondents received also a question, to which they did not respond: how large is this group? How can in 1551 individuals vaccine hesitancy be measured for people who did not receive a booster, since all received a booster? In conclusion, this part of the methods need rephrasing, as it seems to be absolutely confusing. This could be done using a flow chart or separate table.
· Line 194 “After excluding from analysis those who al-ready received a second booster shot, …”. This statement seems to be in contrast with statements later in the text mentioning individuals who received a 2nd booster: see, e.g., line 230 “with 11% [10, 12] of the sample indicating that they have already received an additional second booster”.
· Following this point it should be noted that exclusion of a certain subgroup from the total population creates bias in the interpretation of incidences.
· Line 205 “adjusted”. This term needs explanation: “adjusted for what”? As long as this is not explained, data in Table 1 cannot be interpreted. “Raw” and “Weighted” is not clear nor explanatory.
· The abbreviation “RQ” needs explanation.
· Table 2 needs explanation what is presented. What is “DV” in this table?
· Figure 1 needs explanation. For instance: which group is in this figure, and what is the unit in the y-axis? What means “predicted probability”, line 281?
· Line 353, Informed consent: How was informed consent obtained? Where there individuals who denied informed consent, and how large was this group?
Author Response
REVIEWER 2
This manuscript presents a study on the hesitancy in receiving a 2nd booster vaccination for COVID in
individuals after a 1st booster vaccination. The study was conducted in the USA. The author concludes
in the first sentence of the Discussion “The results presented in this paper suggest that, even among
those Americans who chose to receive a booster vaccine for COVID-19, many would express substantial
reluctance to do so a second time if recommended to do so by public health authorities.”
The topic of this manuscript is indeed quite relevant for a separate study, even considering the intense
discussion in the USA about COVID vaccination. However, the report of this study manifests serious
flaws, both in Materials and Methods, and in the Results. Therefore, it is not possible to have a clear
insight in the study design and hence the results, it is not possible to the reader to understand the
basis on which the conclusion is based. Some points are mentioned below. A serious rephrasing of the
report is indicated.
I thank R2 for both their kind words about this manuscript, as well as the identification of several
areas in which it can be improved. As I detail below in my replies to each of R2’s concerns, I have
made a major effort to substantially rephrase key passages throughout the manuscript; with an
eye toward enhancing clarification of the study’s methods, results, and substantive scope.
This study is limited to the situation in the US: this should be mentioned in the title and emphasized in
the introduction. For instance, the study addresses the 2nd booster which could include components
of the Omicron variant, while in Europe the campaign for a 3rd booster including this variant has been
initiated at the same time.
I have revised the piece’s title to add the phrase “in the U.S.” to clarify the study’s U.S. focus.
Additionally, as R2 helpfully recommends, I have made an expanded effort in the piece’s
Introduction section to make it clear that my focus is exclusively on the United States; including
adding phrases like “in the U.S.” to the piece’s abstract, and replacing instances of the word
“individuals” in the Introduction with “Americans.”
The title needs anyhow a rephrasing: the study does not address “health policy consequences”, and
“social correlates” is a very vague term which raises more questions than answers.
I have rephrased the title in line with R2’s recommendations: removing the term “Social” (as I
also agree that this was both extraneous and unclear) and replacing the term “Health Policy”
with “Public Health” (as this study did not specifically assess policy attitudes).
The author uses the wording “social scientist”. This creates confusion, is the a scientist who is member
of a socialist party, or a scientist who considers him/herself a socialist?
I have removed all instances of the term “social scientist(s)” from this paper, and replaced it with
terms like “researcher(s).” I agree that this language was extraneous and potentially confusing.
For reference: I am a social scientist by training (i.e., with a PhD in a social science field related to
health behavior and promotion).
The phrasings starting with “I”, like I tested, I choose, I analysed, etc, should be largely avoided, as it
insinuates personal opinions instead of a science-based approach.
I have removed all instances (approximately 20) of the active voice first person pronoun “I” from
this manuscript, and replaced them with passive voice references to the study itself.
The introduction is quite long, and lacks relevant information used in vaccination campaigns. For
instance, booster vaccination for older people is to protect themselves, while for younger people it is
to protect the society. Also, the main aim is to avoid hospitalization, and not to get infected with
moderate illness.
I thank R2 for their helpful remarks about the Introduction’s length and substantive content.
Regarding length, I wanted to ensure that I gave a sufficiently detailed nod to past research on
related subjects, without exceeding the journal’s word limits. As this manuscript remains within
the journal’s required limits, and as the other reviewer on this paper did not object to the
Introduction’s length, I opted to keep that section in its present form. However, if there are
specific areas where cutting down that section might help improve readability and/or
comprehension beyond the major changes I’m applying in response to each of R2’s helpful
remarks, I’m happy to consider doing so!
Additionally, I appreciate R2’s recommendation that I focus on differences in booster vaccine
messaging. While I agree that these differences in focus could exist, I’m not aware of any
peer-reviewed research documenting that these differences actually exist in the way the US
health agencies and other communicators are conducting booster vaccination campaigns. In fact,
my colleagues and I are actively studying this very question right now; given what we see as a
dearth of research in this area.
Absent the ability to cite external peer-reviewed sources documenting this claim, I opted not to
include R2’s suggested language in the revised manuscript. I am of course happy to reconsider,
however, if any scholarly work has addressed this question.
Line 145 “Lucid Theorem” should be explained. Apparently individuals selected in this way were
approached with questions. The total of 3950 individuals were limited to 1551 individuals who self
reported a booster vaccination (line 147). This quite unclear, was the study in 1551 or 3950
individuals? Where there people who did not give answers to the questions? See line 153
“Immediately following the presentation of this text, I asked respondents:”. It is assumed that
non-repondents received also a question, to which they did not respond: how large is this group? How
can in 1551 individuals vaccine hesitancy be measured for people who did not receive a booster, since
all received a booster? In conclusion, this part of the methods need rephrasing, as it seems to be
absolutely confusing. This could be done using a flow chart or separate table.
I have made a significant effort in the revised manuscript to clarify the study’s procedures and
data. As R2 rightly notes, I now recognize that the original manuscript failed at times to draw a
distinction between the full study-level sample (N = 3,950; some of whom were not “boosted”)
and the estimation sub-sample of N=1,551 “boosted” Americans, excluding those who had
already received a second booster (I discuss this point in more detail later, in response to R2’s
concerns listed below). I now clarify in the piece’s abstract that most analyses (see rows 1-3 in
Table 1 for additional clarification) are limited to just those N=1,551 Americans who received an
initial booster vaccine, and had not already received a second booster. I have added similar
language to the piece’s Methods section, and noted in Table 2 that multivariate analyses are
limited to this sub-sample of survey respondents.
Additionally, I have also added text clarifying that respondents who were not boosted against
COVID-19 did not receive the question R2 references in line 153. The size of the group not asked
this question is referenced in Table 1, Row 3 (i.e., it is equal to the full study’s valid sample size of
3,950 people minus all “boosted” Americans). Note also that I used this question to identify
whether or not some Americans had already received a second booster (i.e., boosted
respondents could indicate that they had already received the vaccine), meaning that individuals
boosted twice were administered this question prior to being excluded from analysis.
I also want to clarify in this review memo that because this study pertains to Americans’
willingness to receive a second booster vaccination, it is indeed possible to observe variation on
vaccine hesitancy among this sub-sample.
Finally, Although R2 suggests that a Table or Flow Chart might help to clarify these distinctions,
my hope is that – by making several major edits throughout the manuscript’s main text – the
updated piece more clearly identifies who was included in the estimation sub-sample, and how
that group differs from the full survey sample more generally. I am of course happy to continue
to clarify if need be!
Line 194 “After excluding from analysis those who al-ready received a second booster shot, …”. This
statement seems to be in contrast with statements later in the text mentioning individuals who
received a 2nd booster: see, e.g., line 230 “with 11% [10, 12] of the sample indicating that they have
already received an additional second booster”.
I thank R2 for pointing out what appears to be a contradiction in the main text. I have clarified
the main text in line 230 to make clear that the 11% who received a second booster shot are
among those who received a first booster shot, but are excluded from analysis in Table 2. As
noted above, I have also made several edits to the manuscript to clarify that the estimation
sub-sample for most analyses includes individuals who received a first booster shot, and had not
yet received a second.
Following this point it should be noted that exclusion of a certain subgroup from the total population
creates bias in the interpretation of incidences.
I now note directly in the manuscript’s main text that while data from Lucid is demographically
representative of the US adult population on the whole, it is not necessarily representative of
the sub-population of those who received a booster vaccine. That clarification can be found in
the paragraph of text immediately preceding the “Measures” sub-section.
Line 205 “adjusted”. This term needs explanation: “adjusted for what”? As long as this is not
explained, data in Table 1 cannot be interpreted. “Raw” and “Weighted” is not clear nor explanatory.
I have replaced all instances of the term “adjusted” in the main text with the term “weighted;”
i.e., meaning that averages were constructed either with (adjusted) or without (unadjusted) the
application of post-stratification weights, which I describe in the piece’s Methods section. I have
also added analogous text to the note accompanying Table 1.
The abbreviation “RQ” needs explanation.
I now clarify in the main text that RQ stands for “Research Question,” upon the first reference of
this term. I apologize for te confusion regarding the use of the shorthand; this nomenclature is
quite common in my field, but I now recognize may not be used ubiquitously.
Table 2 needs explanation what is presented. What is “DV” in this table?
I have updated Table 2 to clarify that DV stands for “Dependent Variable.”
Figure 1 needs explanation. For instance: which group is in this figure, and what is the unit in the
y-axis? What means “predicted probability”, line 281?
I have clarified the caption of Figure 1 to make clear that only individuals who received a first
(but not a second) booster vaccine are included in the analysis. I have also made clear that the
bars in the figure correspond to predicted probabilities derived from a multivariate logistic
regression model.
Line 353, Informed consent: How was informed consent obtained? Where there individuals who
denied informed consent, and how large was this group?
I have added information to the piece’s informed consent statement noting both how it is that I
obtained informed consent, and describing the size of the group who did not consent to
participate in the study. I also note that non-consenters were escorted to the end of the survey,
and that I collected no data from this group.
Round 2
Reviewer 2 Report
The author has conducted a serious effort in improving the quality of the manuscript, and answering the comments raised by the reviewer. This is highly appreciated. However, there are still major concerns raised in reading the revised manuscript. A major point is the validity of the Lucid database and the process extracting a representative sample for study from this database, i.e., without providing proof of the contrary, “representative” may not apply. The suggestion by the author that the sample is representative is not clear, and this affects the value of the study results. Second, it is still not clear how individuals were selected for further evaluation. Note that this is a main point in every paper dealing with studies on humans: often this aspect is illustrated in a graph. Third, working with uncontrolled self-reports creates bias, and hence affects the value of the design with regard to sending out questions to people in study. Fourth, the use of unweighted and weighted data is still fully unclear to the reader. Fifth, the data presentation is unclear, as it mainly describes the statistical methods and not the actually measured data. Finally, the Introduction and Discussion are by far too large, and are in part beyond the actual study and study data, or even an overinterpretation of data, also considering the fact that individuals who were likely to go for follow-up boosters were not studied in detail.
· This reviewer still does not know what Lucid is: maybe this is well known by social scientists, it is not so in the biomedical arena. Is this a reliable officially validated database, or is this fake? Apparently, Lucid measures vaccination attitudes, see Discussion, which might be different from taking or rejecting a vaccination: what does attitude mean, and how is attitude measured? This question is relevant, as the attitude associates likelihood to take a vaccination but is more as it is linked with behavior and expression of opinion. What are the data on which the author concludes the quality of this database regarding demographic and other parameters, like party membership, voting behavior, religion, location of living, income, etc etc, all factors that are associated with likelihood to get vaccinated (there are many more factors)? Knowing that such factors exists, it might not be allowed to combine subgroups according to these factors in one group, including presentation of averages etc: not presenting subgroups has consequence regarding the value of the study in terms of “Correlates” and “Public Health” in the title.
· Assuming that Lucid has more entries than 3950 adults, it is not described how the selection of a “representative” sample was done: this very relevant to understand the value of the report. The author mentions “formally representative data” in the Discussion: does this mean that Lucid comprises non-formally representative data? There is no check whether the self-reported conditions in Lucid were actually true: self-reports have the intrinsic disadvantage of being biased.
· Hence, the sentence “Correspondingly, although the sampling service is neither formally representative of the US population nor (necessarily) the subgroup of “boosted” Americans referenced above, it is nevertheless well-suited for studying the research questions tested in this piece.” is killing with respect to the validity of the report. “well-suited” may just represent the personal opinion of the author and not general consensus. Why is this considered “well-suited”?
· Weighting: still quite unclear and not explained-in-full: this reader does not understand this activity at all: one reason is that factors used in weighing are not mentioned in the main text. Maybe, this is described in supplementary material, but if so, this is not considered pivotal by the author to inform the reader. For a not-involved reader the statement “Adjusted averages are calculated via the application of post-stratification weights.” is not understandable.
· There are no data on the response rate of additional questions. Did all people who were invited to ask a question actually respond? If not (which seems logical since there was no pressure to answer), what are the differences between respondents and non-respondents? Note, this point is different from that on informed consent mentioned below.
· It might be interesting to evaluate the differences in composition (demographics etc, see above) between the group that likely will take a booster and the group that likely will not take a booster. Considering the title of the report, especially the wording “Correlates” and “Public Health” this would be a logical evaluation to conduct. This evaluation may include asking similar (evidently not the same) questions to understand why people are likely to take a booster, as the questions send to the hesitant people. A rationale should be given why the group of people who likely will take a booster, has not been investigated in further detail.
· To follow the previous point: in the Discussion the author writes “These findings have important health policy implications.” The report only describes some aspects regarding hesitancy in those who unlikely would take a booster. Reasons to take a booster are not investigated, and this might be of the same or even higher importance, considering that the respective group is the majority in the total. This needs an explanation.
· In data presentation, the author does not clearly describe what is presented, as legends to tables and figures mainly mention the statistical procedures. This in particular applies to table 2 and figure 1.
· Informed consent: it is mentioned that 205 people were excluded because they did not consent. This is a large number with respect to the number of people who did participate (n=1551), and obviously affects representability. This needs to be mentioned as a limitation in considering the validity of the report.
Author Response
Dear Editors at Vaccines and Anonymous Peer Reviewer #2,
Once again, I’d like to sincerely thank you for taking the time to review my manuscript entitled The Correlates and Public Health Consequences of Prospective Vaccine Hesitancy Among Individuals who Received COVID-19 Vaccine Boosters in the U.S . I was delighted to hear that R1 was satisfied with the changes I made to the original manuscript, and that the piece received a revise decision at this journal.
I especially appreciate R2’s willingness to read the manuscript for a second time, and offer detailed feedback regarding how it might be improved going forward. I have made several significant changes to the manuscript’s main text, in response to each of R2’s remaining concerns.
As was the case in the previous round of reviews, I have re-printed each of R2’s comments (bolded), and listed my reply to each one directly below. Where relevant, I now directly quote from the manuscript (in serif font) to highlight where I have made revisions (bolded, serif font).
I hope that these changes demonstrate that I both appreciate R2’s remaining concerns, and that I have taken them very seriously in revising this piece. I again want to thank both the Reviewer and Editors for their consideration, and look forward to working with you throughout the peer review process.
Sincerely,
- Matt
REVIEWER 2
The author has conducted a serious effort in improving the quality of the manuscript, and answering the comments raised by the reviewer. This is highly appreciated. However, there are still major concerns raised in reading the revised manuscript. A major point is the validity of the Lucid database and the process extracting a representative sample for study from this database, i.e., without providing proof of the contrary, “representative” may not apply. The suggestion by the author that the sample is representative is not clear, and this affects the value of the study results. Second, it is still not clear how individuals were selected for further evaluation. Note that this is a main point in every paper dealing with studies on humans: often this aspect is illustrated in a graph. Third, working with uncontrolled self-reports creates bias, and hence affects the value of the design with regard to sending out questions to people in study. Fourth, the use of unweighted and weighted data is still fully unclear to the reader. Fifth, the data presentation is unclear, as it mainly describes the statistical methods and not the actually measured data. Finally, the Introduction and Discussion are by far too large, and are in part beyond the actual study and study data, or even an overinterpretation of data, also considering the fact that individuals who were likely to go for follow-up boosters were not studied in detail.
I thank R2 for their kind words about the revised manuscript, and their willingness to carefully evaluate this manuscript for a second time. I also appreciate their identification of several ways in which I can continue to improve the quality of this manuscript. I have responded to each of the Reviewer’s remaining concerns in the pages that follow, and detail how I have updated the manuscript accordingly.
This reviewer still does not know what Lucid is: maybe this is well known by social scientists, it is not so in the biomedical arena. Is this a reliable officially validated database, or is this fake? Apparently, Lucid measures vaccination attitudes, see Discussion, which might be different from taking or rejecting a vaccination: what does attitude mean, and how is attitude measured? This question is relevant, as the attitude associates likelihood to take a vaccination but is more as it is linked with behavior and expression of opinion. What are the data on which the author concludes the quality of this database regarding demographic and other parameters, like party membership, voting behavior, religion, location of living, income, etc etc, all factors that are associated with likelihood to get vaccinated (there are many more factors)? Knowing that such factors exists, it might not be allowed to combine subgroups according to these factors in one group, including presentation of averages etc: not presenting subgroups has consequence regarding the value of the study in terms of “Correlates” and “Public Health” in the title.
I now recognize that even my revised description of Lucid Theorem – which is an online, opt-in panel sampling service – could have done more to present readers with important information about what it is, and how it operates. As R2 rightly notes, Lucid is indeed widely used in the social sciences, and is likely familiar to most social science researchers. Consequently, I recognize that it is important that I describe the service in much greater detail, as to facilitate comprehension beyond the social science community.
To this end, I have made several major changes to the manuscript. I detail several of these changes in the pages that follow. Most relevant to the above concern is the addition of the following paragraph, wherein I provide additional information about the service; citing relevant research documenting the panel’s size and demographic representativeness. I also describe the service’s quota sampling procedures in greater detail.
That paragraph reads as follows:
Lucid Theorem is an online opt-in sampling service that retains a large panel of potential survey respondents (over 375,000 unique daily users are available for contact, by some estimates). [19] The service invites respondents to participate in surveys using quota sampling methods. This means that, based on responses to an initial demographic inventory survey, the service selectively sends survey-taking opportunities to potential respondents who fill quotas derived from nationally representative demographic benchmarks obtained from the US Census.
More informally, I also want to note in this review memo that online opt-in panel service data is employed widely in the study of vaccine hesitancy (as I mentioned in the original manuscript), which includes studies published in this journal (for example, this recent paper: https://www.mdpi.com/2076-393X/9/4/346/htm).
Finally, I have supplemented my discussion recognizing the limitations of self-reported online survey data in assessing vaccine hesitancy. While I had previously included language documenting high levels of similarity between self-reports and actual behavior, I recognize that this correspondence is not perfect. Thus, I have clarified in the Discussion section that future efforts to study related topics ought to make an effort – where possible – to validate self-reports by linking medical records to survey data.
Specifically, I write:
Of course, these findings are not without limitations. First, although past research suggests that vaccination attitudes and intentions tend to strongly predict actual behavior [31-33], prospective behavioral self-reports do not necessarily guarantee that individuals will (or will not) choose to vaccinate. Moreover, these data are derived from a single online, opt-in cross-sectional survey. Although vaccine attitudes measured in Lucid data have been shown to closely mirror more-representative surveys [20], panelists may hold attitudes or express behaviors that differ from the US population in potentially-unobserved ways. Efforts to replicate this work in formally representative data and/or to validate vaccination self-reports with verified medical records thereby offer an important avenue for future research.
Assuming that Lucid has more entries than 3950 adults, it is not described how the selection of a “representative” sample was done: this very relevant to understand the value of the report. The author mentions “formally representative data” in the Discussion: does this mean that Lucid comprises non-formally representative data? There is no check whether the self-reported conditions in Lucid were actually true: self-reports have the intrinsic disadvantage of being biased.
As I now clarify above in response to R2’s earlier point, Lucid’s sampling service does indeed include more than 3,950 respondents. In fact, over 350,000 unique users on a daily basis. I clarify this point in the passage quoted below.
I also want to clarify that when I say that Lucid’s data are not formally nationally representative, I mean that they are not a true probability sample of the American public (very few public opinion studies can claim to be true probability samples). However, Lucid data are what survey researchers call demographically representative; meaning that, while they are not formal probability samples of the American public, they nevertheless reflect nationally representative Census benchmarks on a series of observable factors. As I noted in the original manuscript, demographically representative data from Lucid have been shown to produce estimates of vaccine attitudes and behaviors that closely mirror nationally representative benchmarks.
Still, I realize that my discussion of demographic representativeness could have been much clearer. To this end, I have added the following text to the paragraph referenced in R2’s above comments. I have also migrated my discussion of the panel service’s suitability for studying vaccine hesitancy to this paragraph (more on changes to that language in a moment, in response to R2’s next set of concerns).
That text reads as follows:
Although data from Lucid are not formally nationally representative, the service’s quota sampling procedures to recruit participants from its large, online, opt-in sampling frame that have been shown to build samples that are demographically representative of US adults with respect to respondents’ resemble age, gender identity, income, racial identity, political party, and geographic region benchmarks. [19] Correspondingly, data from Lucid have been shown to produce estimates of vaccine hesitancy that mirror those obtained in nationally representative probability samples [20]. [...]
Hence, the sentence “Correspondingly, although the sampling service is neither formally representative of the US population nor (necessarily) the subgroup of “boosted” Americans referenced above, it is nevertheless well-suited for studying the research questions tested in this piece.” is killing with respect to the validity of the report. “well-suited” may just represent the personal opinion of the author and not general consensus. Why is this considered “well-suited”?
I also recognize that my initial discussion of why Lucid data are well-suited for the study of vaccine hesitancy was incomplete. I have supplemented my discussion of this point by (a) migrating my discussion of “suitability” to directly follow the revised passage discussing the service’s demographic representativeness (as noted above), and (b) made clearer why the service’s widespread usage in studying similar questions and demographic representativeness make it well-suited for use in this research.
Specifically, I write:
[...] Correspondingly, data from Lucid have been shown to produce estimates of vaccine hesitancy that mirror those obtained in nationally representative probability samples [20]. Data from Lucid have also been shown to replicate well-studied experimental treatment effects both prior to and throughout the COVID-19 pandemic [21]; tend to better reflect US population demographic benchmarks than traditional convenience samples [22]; and have been employed extensively in the study of COVID-19 vaccine attitudes and behaviors in the US [19, 23-25]. Correspondingly, although the sampling service is neither formally representative of the US population nor (necessarily) the subgroup of “boosted” Americans referenced above, Lucid’s ability to produce large and demographically representative samples (on the aforementioned factors) – coupled with its widespread usage and validation in the study of vaccine attitudes and behavior – nevertheless make it well-suited for studying the research questions I tested in this piece.
Weighting: still quite unclear and not explained-in-full: this reader does not understand this activity at all: one reason is that factors used in weighing are not mentioned in the main text. Maybe, this is described in supplementary material, but if so, this is not considered pivotal by the author to inform the reader. For a not-involved reader the statement “Adjusted averages are calculated via the application of post-stratification weights.” is not understandable.
I also appreciate R2’s willingness to draw attention to potential ambiguities concerning the post-stratification procedure. These methods are common in social scientific studies of vaccine hesitancy, but I recognize that they may be unfamiliar to those in other disciplines. I have therefore added the following language to the main text to clarify how this procedure operates:
To account for any potential remaining deviations between the resulting sample and the US population, I calculate post-stratification weights that adjust the data to Census benchmarks – such that under-represented responses relative to national benchmarks are treated as more influential (“up-weighted”) and over-represented respondents are treated as less influential (“down-weighted”) in the estimation of weighted means – were calculated on respondents’ age, income, gender identity, and racial identity [2019].
I also want to point out here that all quantities listed in Table 1 are calculated both with and without the application of survey weights. As the table shows, these quantities tend to be quite similar. I therefore hope to have provided readers with a pluralistic set of estimates that they can draw on when interpreting the study’s key results (i.e., as some may prefer the weighted quantities; others, the unweighted quantities), while nevertheless recognizing that this analytical move does not substantively impact the conclusions drawn from this piece.
There are no data on the response rate of additional questions. Did all people who were invited to ask a question actually respond? If not (which seems logical since there was no pressure to answer), what are the differences between respondents and non-respondents? Note, this point is different from that on informed consent mentioned below.
I appreciate R2’s willingness to draw attention to both item nonresponse (i.e., the response rate for specific questions) as well as unit nonresponse (i.e., response rate for the survey itself).
Dealing first with the former, per the comments listed above, I now clarify that all respondents answered the questions pertaining to both their vaccination status and reasons for not vaccinating. As I now note in the main text, respondents were not technically required to answer every question in the survey. However, they received a prompt asking them to fill out unanswered questions before advancing, if they happened to leave a question blank. This procedure resulted in a response rate of 100% for those individual survey items.
I now write:
Note also that, for analytical completeness, all results are presented both with and without the application of survey weights. A comparison of weighted and unweighted sample statistics to nationally representative benchmarks is available in the Supplementary Materials. Additionally, all respondents were prevented from leaving questions blank in the survey without first reading a short (automatically generated) reminder to fill out unanswered questions. Correspondingly, there was no missing data on any of the variables featured in Table 1.
It might be interesting to evaluate the differences in composition (demographics etc, see above) between the group that likely will take a booster and the group that likely will not take a booster. Considering the title of the report, especially the wording “Correlates” and “Public Health” this would be a logical evaluation to conduct. This evaluation may include asking similar (evidently not the same) questions to understand why people are likely to take a booster, as the questions send to the hesitant people. A rationale should be given why the group of people who likely will take a booster, has not been investigated in further detail.
I completely agree with R2’s assessment that studying demographic asymmetries in vaccine uptake is a useful area for research. In fact, I wanted to point out that Table 2 already includes some of these analyses. I.e., by including control variables that account for the possibility that some demographic factors might alternatively explain why some people opt to vaccinate while others do not, these analyses document potential demographic asymmetries between groups.
I mention these quantities only in passing in the main text, as this topic has received substantial attention in previous research on vaccine hesitancy more generally (i.e., both prior to and throughout the COVID-19 pandemic; e.g., citations 11, 20-21, 24-25, 27). Given the relatively short format requested by the journal’s Communication style guide, I opted to focus on the aspects of this study that were more novel. Still, I agree in principle with the reviewer on this score and appreciate their helpful feedback.
To follow the previous point: in the Discussion the author writes “These findings have important health policy implications.” The report only describes some aspects regarding hesitancy in those who unlikely would take a booster. Reasons to take a booster are not investigated, and this might be of the same or even higher importance, considering that the respective group is the majority in the total. This needs an explanation.
As R2 rightly notes, I had originally intended to strike all references to “health policy implications” in the previous round of review. The passage referenced above was an oversight. I apologize for that error! I have subsequently removed the phrase, and replaced it with an more accurate description of the study’s contributions; i.e., its focus on the public health consequences of vaccine hesitancy.
In data presentation, the author does not clearly describe what is presented, as legends to tables and figures mainly mention the statistical procedures. This in particular applies to table 2 and figure 1.
I appreciate R2’s attentiveness to detail when noting ambiguity in the discussion of the results presented in Tables 1 and 2. I now recognize that I could have done more to better introduce both sets of analyses, and discuss exactly what quantities were presented in each one.
To this end, I have added the following language preceding my discussion of the results presented in Table 2:
Table 2 expands on these analyses by presenting the results of the previously-mentioned multivariate ordered logistic regression models. Parameter estimates are listed at the top of each cell, with standard errors in parentheses. The results presented in the table suggest that – controlling for a variety of other factors thought to be associated with COVID-19 vaccine hesitancy – only concern about missing work to get vaccinated (B = 0.54, p = 0.05; two-tailed) and lack of concern about the need for a second booster (B = 2.25, p < 0.01) are significantly associated with intentions to refuse a second booster vaccine.
I have also added the following language preceding the results presented in Figure 1:
Recognizing that ordered logistic parameter estimates are not easily interpretable on their own, Figure 1 uses the information presented in Table 2 to calculate the predicted probability that respondents are “very likely” to receive a second booster. Probing just the statistically significant terms, and holding all other covariates at their sample means, the predicted probabilities presented in Figure 1 suggest that concerns about missing work to get vaccinated is associated with a 10 percentage point decrease in the predicted likelihood that respondents are “very likely” to plan to receive a second booster (from 66 to 56%); a difference which is statistically significant at the p < 0.05 level, two-tailed (t = -2.03; p = 0.04). Similarly, albeit at a much greater magnitude, a lack of concern about the need to boost a second time is associated with a 47% decrease in the likelihood of being “very likely” to do so (from 77 to 30%). These quantities too are significantly different from one another at the p < 0.05 level, two-tailed (t = -22.62, p < 0.01).
My hope is that this additional language helps the reader to more-easily interpret the analyses that follow, and facilitates comprehension of the piece’s main findings.
Informed consent: it is mentioned that 205 people were excluded because they did not consent. This is a large number with respect to the number of people who did participate (n=1551), and obviously affects representability. This needs to be mentioned as a limitation in considering the validity of the report.
As alluded to above, I have now added language to the manuscript’s main text that includes the survey’s completion rate. Before sharing that language, I want to first clarify that the 205 individuals who did not complete the survey (“unit nonresponse”) are based on all 3,950 individuals who took part in the study. This means that the compliance rate was actually quite high; in excess of 95%.
Correspondingly, I have added language to the piece’s Data section that clarifies this point:
Data. Data for this study are derived from a demographically representative sample of N = 3,950 US adults, conducted between April 22 - 27, 2022 via Lucid Theorem. Lucid invited 4,155 individuals to participate in this study, and 205 did not complete the survey. This is indicative of a 95% compliance rate.
Round 3
Reviewer 2 Report
The author has considered the comments on the revised manuscript and has given detailed responses to the concerns expressed in the review. Also some relevant text has been added to the text. This all is highly appreciated by the reviewer.